# Real-time estimation of disease activity in emerging outbreaks using internet search information

**Emily L. Aiken** [1] *, **Sarah F. McGough** [2], **Maimuna S. Majumder** [3], **Gal Wachtel** [4], **Andre T. Nguyen** [5,6], **Cecile Viboud** [7], **Mauricio Santillana** [1,4,8] *

**1** School of Engineering and Applied Sciences, Harvard University, Cambridge, Massachusetts, United States of America, **2** Harvard T.H. Chan School of Public Health, Boston, Massachusetts, United States of America, **3** Department of Healthcare Policy, Harvard Medical School, Boston, Massachusetts, United States of America, **4** Computational Health Informatics Program, Boston Children's Hospital, Boston, Massachusetts, United States of America, **5** Booz Allen Hamilton, Columbia, Maryland, United States of America, **6** University of Maryland, Baltimore County, Baltimore, Maryland, United States of America, **7** Fogarty International Center, National Institutes of Health, Bethesda, Maryland, United States of America, **8** Department of Pediatrics, Harvard Medical School, Boston, Massachusetts, United States of America

* emilyaiken@berkeley.edu (ELA); msantill@g.harvard.edu (MS)

**Data Availability Statement:** All models and evaluation metrics are implemented in Python 3.6 with scikit-learn 0.19.1. All scripts and data used in

## Abstract

Understanding the behavior of emerging disease outbreaks in, or ahead of, real-time could help healthcare officials better design interventions to mitigate impacts on affected populations. Most healthcare-based disease surveillance systems, however, have significant inherent reporting delays due to data collection, aggregation, and distribution processes. Recent work has shown that machine learning methods leveraging a combination of traditionally collected epidemiological information and novel Internet-based data sources, such as disease-related Internet search activity, can produce meaningful "nowcasts" of disease incidence ahead of healthcare-based estimates, with most successful case studies focusing on endemic and seasonal diseases such as influenza and dengue. Here, we apply similar computational methods to emerging outbreaks in geographic regions where no historical presence of the disease of interest has been observed. By combining limited available historical epidemiological data available with disease-related Internet search activity, we retrospectively estimate disease activity in five recent outbreaks weeks ahead of traditional surveillance methods. We find that the proposed computational methods frequently provide useful real-time incidence estimates that can help fill temporal data gaps resulting from surveillance reporting delays. However, the proposed methods are limited by issues of sample bias and skew in search query volumes, perhaps as a result of media coverage.

## Author summary

Public health officials regularly make choices about treatment and prevention in disease outbreaks that have the potential to impact entire affected populations. Often these decisions are based on incomplete or unreliable information due to inherent reporting delays

this study are publicly available at https://github.com/emilylaiken/outbreak-nowcasting.

**Funding:** MS was funded in part by the Bill and Melinda Gates Foundation (OPP 1195154, https://www.gatesfoundation.org/). MS was funded in part by the National Institute of General Medical Sciences of the National Institutes of Health (Award Number R01GM130668, https://www.nigms.nih.gov/). The funders had no role in study design, data collection and analysis, decision to publish, or preparation of the manuscript.

**Competing interests:** The authors have declared that no competing interests exist.

in healthcare-based disease surveillance systems. This issue of public health decision-making based on limited data is even more salient in emerging outbreaks, which are typically characterized by uncertain disease dynamics and limited surveillance capacity. We demonstrate the potential for using digital trace data—in this case, Internet-based information from Google search trends—for estimating disease activity in emerging outbreaks in the absence of accurate real-time healthcare-based data sources. We evaluate how data-driven methods leveraging search trend data would have performed in real-time in five recent outbreaks (yellow fever in Angola, Zika in Colombia, Ebola in the DRC, plague in Madagascar, and cholera in Yemen), and find that the methods frequently provide useful signals of disease activity ahead of standard healthcare-based surveillance methods.

## Introduction

Disease outbreaks have been major drivers of morbidity and mortality since the beginning of recorded history and continue to pose a major threat to humankind. Surveillance of disease outbreaks by healthcare systems is key to effective outbreak response. In particular, surveillance data is necessary to determine the overall scale of response to an outbreak, allocate limited resources for treatment and prevention, and effectively time interventions to minimize impacts [1]. Epidemiologists use surveillance data to estimate important features of an outbreak, such as morbidity and mortality burden, case fatality rate, and transmission patterns. In recent years, the use of mathematical modeling of disease activity and transmission to predict the likely trajectory of an outbreak and guide intervention strategies has been increasingly explored [1–4].

It is particularly challenging to monitor and characterize unexpected (emerging) disease outbreaks in regions that have not experienced the presence of a specific pathogen in recent times. Such emerging disease outbreaks, especially in their early stages, are characterized by incomplete, delayed, and biased epidemiological surveillance data [1]. Reporting delays in surveillance systems inevitably emerge from limited healthcare resources and coverage, as well as the time required to process lab tests and clean, anonymize, aggregate, and communicate data from distributed healthcare facilities to central authorities. These reporting delays and issues of missingness are manifested in epidemiological reports released by the World Health Organization (WHO) and other health authorities for several recent outbreaks [5–11].

Novel Internet-based data sources have the potential to fill some of these temporal "data gaps" in tracking emerging outbreaks. Research to date on using Internet-based data sources to provide early estimations of disease activity has shown promising results for endemic diseases in high- and middle-income countries, including influenza in the United States [12–17] and dengue in Brazil, Mexico, Thailand, Singapore, and Taiwan [18]. Digital epidemiological methods use mathematical methods to combine Internet-based data—including Google search trends (data on aggregated Google query volumes) [12, 13, 18], Twitter microblogs [14, 15, 19], online news aggregators [20], electronic medical records [21, 22], and crowdsourced disease activity estimates [23, 24]—with historic epidemiological data to produce real-time estimates of disease activity ("nowcasts").

The most famous digital epidemiological study to date, Google Flu Trends [12], tracked national influenza rates in the United States using Google query volumes and autoregressive epidemiological data. It was famously discontinued in 2013 after underestimating the H1N1 outbreak of 2009 and missing by large margins in subsequent influenza seasons [25]. Google Dengue Trends, a related project tracking dengue fever using Internet query data in Bolivia,

Brazil, India, Indonesia, and Singapore [26] was similarly discontinued in August 2015 [18]. In 2015, researchers revised the Google Flu Trends algorithm for tracking seasonal and endemic diseases as ARGO, a machine learning approach based on a dynamic multivariate regularized regression that leverages autoregressive epidemiological data along with real-time generalized online data sources, including Google search trends, Twitter microblogs, electronic health records, and others [13, 27]. ARGO has been shown to produce meaningful and accurate national-level disease activity estimates for influenza in the US and Latin America, and dengue in several middle income countries, weeks ahead of reports issued by traditional surveillance systems [13, 17, 18].

Adapting digital epidemiological methods like ARGO for tracking emerging outbreaks in developing regions brings up a host of new challenges relating to an absence of historical epidemiological data for training and validation, as well as a paucity of digital data due to poorer Internet coverage. To our knowledge, three past studies have experimented with Internet-based data for emerging infections: Majumder et al. [28] demonstrate the use of digital data sources (including Google search trends and news reports) to provide estimates of $R_0$, the basic reproductive number, in the absence of real-time epidemiological surveillance data in the 2016 Latin American Zika outbreak. Chunara et al. [29] use Twitter and news report data to estimate $R_0$ in the 2010 Haitian cholera outbreak. In the only work to date on nowcasting disease incidence in an emerging outbreak with digital data sources, McGough et al. [30] incorporate information from Google search trends, Twitter, and news reports to produce nowcasts of incidence in the 2015-2016 Latin American Zika outbreak 1-3 weeks in advance of standard epidemiological reports.

## Our contribution

Here we expand on McGough et al. to evaluate the performance of digital epidemiological methods for nowcasting five contemporary outbreaks: Yellow fever in Angola (2016), Zika in Colombia (2015-2016), Ebola in the Democratic Republic of the Congo (2018-present), pneumonic plague in Madagascar (2017), and cholera in Yemen (2016-2017). We propose three simple data-driven predictive models: a linear autoregression that uses historic epidemiological data to produce real-time disease activity estimates (AR), a linear regression that leverages observed Google query volumes to estimate disease incidence (GT), and a regression on both historic epidemiological data and search query data (ARGO). We find that ARGO provides useful estimates of disease activity for yellow fever in Angola, Zika in Colombia, and plague in Madagascar weeks earlier than traditional healthcare-based surveillance data. We find that our data-driven methods are less effective at tracking Ebola in the DRC and cholera in Yemen, and hypothesize that issues of sample bias and skew in search query volumes as a result of media coverage may contribute to a poor signal in these cases.

## Results

### Motivation for digital epidemiological methods

To motivate the use of digital data streams to monitor emerging outbreaks, we produced a series of correlations assessing the relationship between each outbreak's epidemiological curve and the volume of a simple Google search term querying the disease of interest (e.g. the search term "Zika" in the case of Colombia). As shown in Fig 1, the search volumes appear to track the time series of cases synchronously in most countries, and we observed high correlations for Angola (r = 0.84, yellow fever), Colombia (r = 0.80, Zika), and Madagascar (r = 0.73, plague), suggesting the potential utility of digital data-driven epidemiological models.

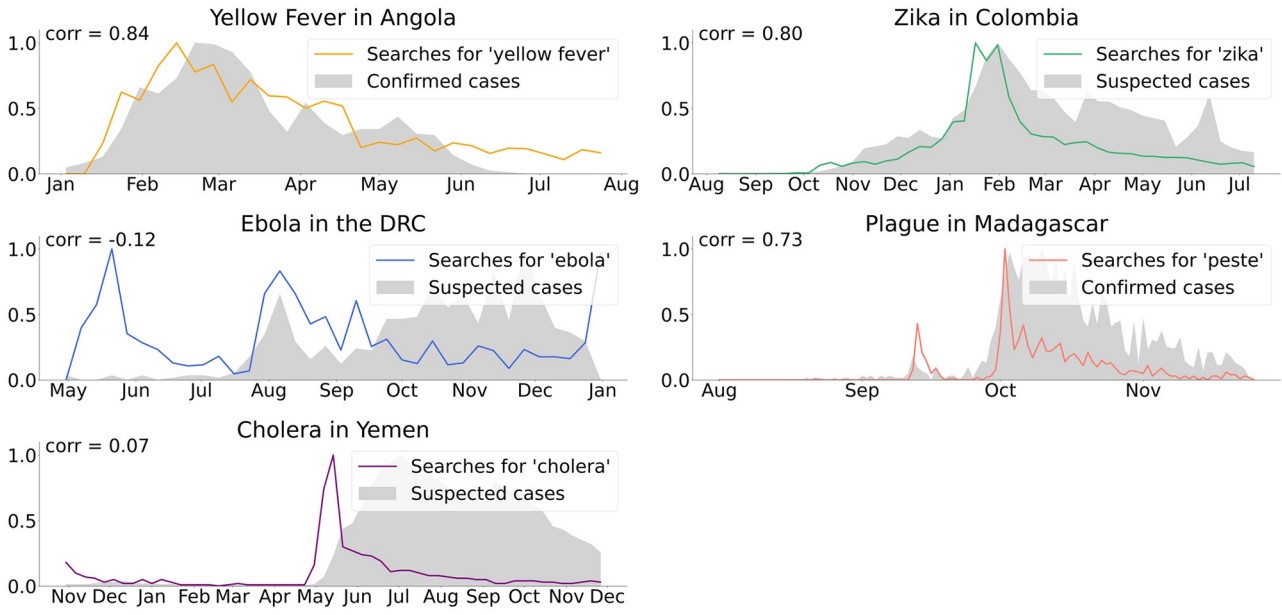

**Fig 1. Motivation for digital epidemiological modeling of five emerging outbreaks.** In each case, the outbreak's epidemiological curve (in grey, normalized to the range [0, 1]) is compared with normalized search volumes for a single related search term within the country in question.

For each disease outbreak, we built three machine learning models to produce (retrospective and out-of-sample) real-time disease activity estimates that use input information that would have been available at the time of prediction. Our three models were trained dynamically on a continuously expanding time window to incorporate new information as it became available and are summarized as follows: (1) Autoregressive model (AR), that uses only historical cases from $n$ weeks in the past to predict current cases; (2) Google search trends (GT), a multivariate model that uses only synchronous Google search terms for prediction; and (3) ARGO, a multivariate model similar to the one presented in [13] that combines both autoregressive case information and Google searches to make predictions. A handful of simple search terms were selected for inclusion in the GT and ARGO models based on their obvious relevance to the disease in question. We assessed the predictive performance of each model when compared to subsequent observations by healthcare-based disease surveillance systems. Details of model implementation can be found in the Materials and Methods section.

## Evaluation assuming continuous flow of available epidemiological data

As a reality check, our first series of models compared nowcasts 1- and 2-weeks ahead of the release of case reports with the ground truth incidence available retrospectively in weekly epidemiological updates produced by local health authorities. These models were trained and built with a strategy similar to the one used in endemic and seasonal outbreaks to make sure our efforts could produce meaningful disease estimates under the assumption that disease activity reports become available with delays of one to two weeks and are continuously available. This assumption is not always satisfied in emerging disease outbreaks. Fig 2 shows these predictions over the full time series of each outbreak, while Table 1 summarizes the out-of-sample predictive performance across models and countries as captured by Pearson's correlation (CORR), root-mean-square error (RMSE), and relative root-mean-square error (rRMSE).

We found that, based on RMSE and correlation, digital epidemiological models that incorporated Google information (GT and ARGO) led to reasonable disease estimates that were

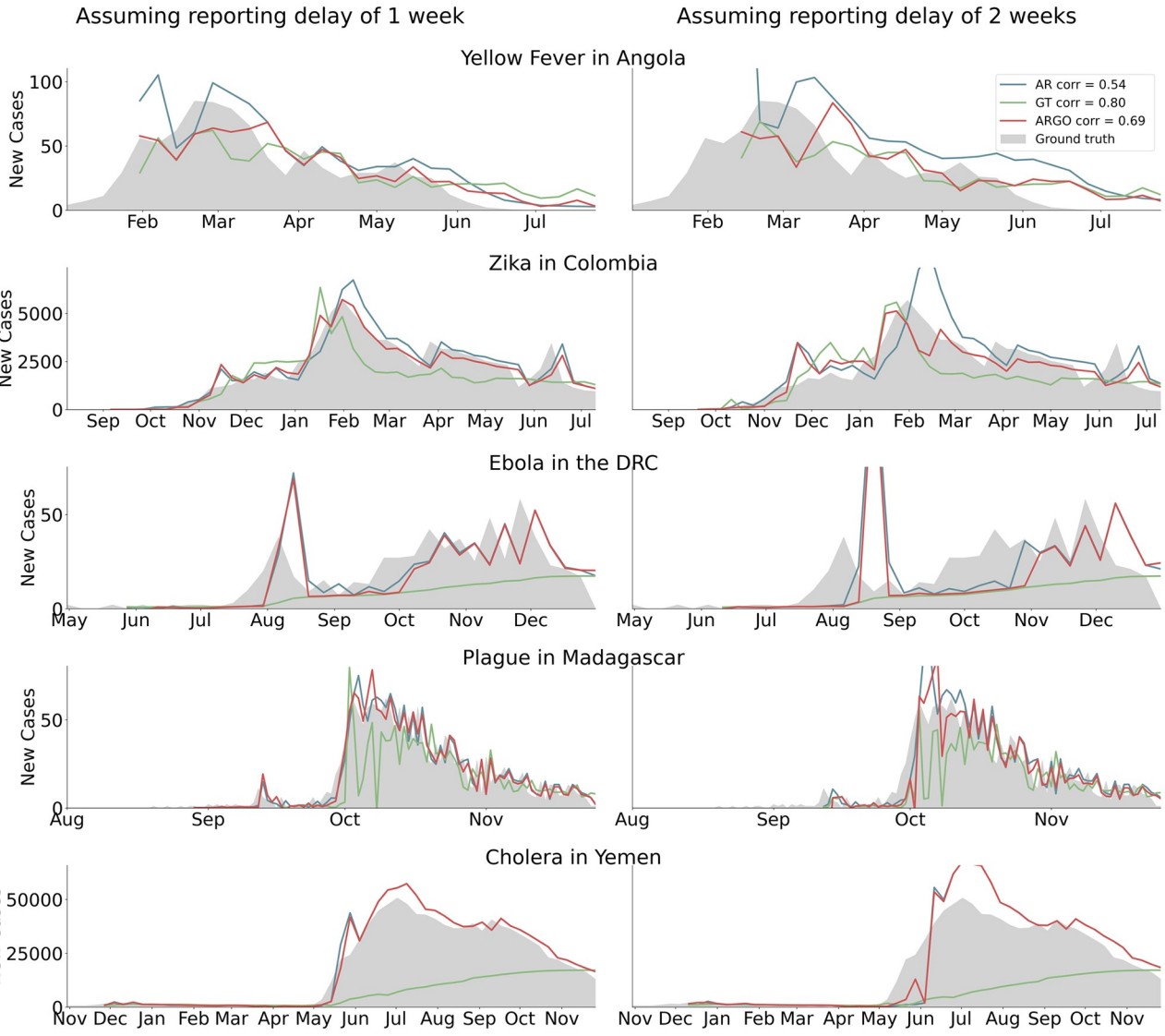

**Fig 2. Series of plots comparing the nowcasts produced by three digital epidemiological models (available in real-time) to "ground truth" epidemiological data (available at a delay).** The left column shows how models perform assuming a 1-week reporting delay in the traditional surveillance system; the right columns shows model performance assuming a 2-week reporting delay.

within range of the observed disease activity. Specifically, GT and ARGO outperformed a naïve autoregressive approach (AR) in all outbreaks and prediction horizons besides plague, in which a pure AR model performed best for 2-week delays. In general, ARGO exhibited the lowest RMSE and highest correlation in a majority of countries and prediction horizons, though Google data alone improved predictions in the case of 2-week delays in two of the outbreaks (yellow fever and Ebola). We note, however, that nowcast models were generally not skillful enough to track Ebola in the DRC, which exhibited substantially lower predictive performance compared to the other countries (correlation range: 0.17-0.58). Moreover, we observed that the ARGO method does not improve significantly upon a naïve autoregressive approach for tracking either Ebola in the DRC or Cholera in Yemen.

To assess the predictive power of the Google search terms used to nowcast cases each week and visualize changes in predictive power over the course of the epidemic, the size of ARGO

**Table 1. Evaluations of three computational models (AR, GT, and ARGO) across five outbreaks, based on correlation ($r$, Panel A), root-mean-square error (RMSE, Panel B), and relative root-mean-square error (rRMSE, Panel C).** The result of the best-performing model for each prediction scenario and metric is bolded. It is important to note that the units of the error (RMSE) are different given that the magnitude of each outbreak was different. The relative error, however, is comparable across outbreaks.

| Delay (weeks) | Yellow Fever | | Zika | | Ebola | | Plague | | Cholera | |
|---|---|---|---|---|---|---|---|---|---|---|
| | **1** | **2** | **1** | **2** | **1** | **2** | **1** | **2** | **1** | **2** |
| *Correlation (r)* | | | | | | | | | | |
| AR | 0.879 | 0.54 | 0.92 | 0.78 | 0.57 | 0.19 | 0.91 | **0.88** | 0.98 | 0.93 |
| GT | 0.79 | **0.80** | 0.78 | 0.73 | **0.582** | **0.50** | 0.74 | 0.68 | 0.65 | 0.59 |
| ARGO | **0.882** | 0.69 | **0.93** | **0.82** | 0.581 | 0.17 | **0.92** | 0.84 | **0.99** | **0.94** |
| *Root-mean-square error (RMSE)* | | | | | | | | | | |
| AR | 17.60 | 62.65 | 644.24 | 1176.74 | 15.252 | 28.11 | 8.45 | **11.65** | 4224.88 | 9156.57 |
| GT | 17.66 | **17.63** | 997.45 | 1072.01 | 16.98 | **18.13** | 13.60 | 15.38 | 18532.22 | 19486.67 |
| ARGO | **13.22** | 20.42 | **542.39** | **823.34** | **15.246** | 27.41 | **7.97** | 11.85 | **3973.06** | **8497.43** |
| *Relative root-mean-square error (rRMSE)* | | | | | | | | | | |
| AR | 0.55 | 2.10 | 0.31 | 0.54 | **0.81** | 1.40 | 0.45 | **0.53** | 0.23 | 0.48 |
| GT | 0.56 | **0.59** | 0.58 | 0.50 | 0.90 | **0.90** | .72 | 0.70 | 1.01 | 1.03 |
| ARGO | **0.42** | 0.69 | **0.26** | **0.38** | **0.81** | 1.37 | **0.42** | 0.54 | **0.22** | **0.44** |

model coefficients for each week of prediction are shown for each country in Fig 3, S1–S5 Figs. Because the models are dynamically trained on a 1-week expanding time window, the predictive power of the variables are seen to fluctuate over the weeks of the outbreak, with many search terms appearing most important for prediction in early stages of the outbreak.

## Evaluation based on publicly released reports

The first evaluation approach assumed that the ground truth (weekly cases) were reported accurately within 1-2 weeks of their occurrence, which is rarely the case in emerging outbreaks in which surveillance may be constrained by limited resources.

In our second approach, we evaluated the performance of the same three models (AR, GT, and ARGO) under more realistic conditions, using partial and unrevised case reports as they

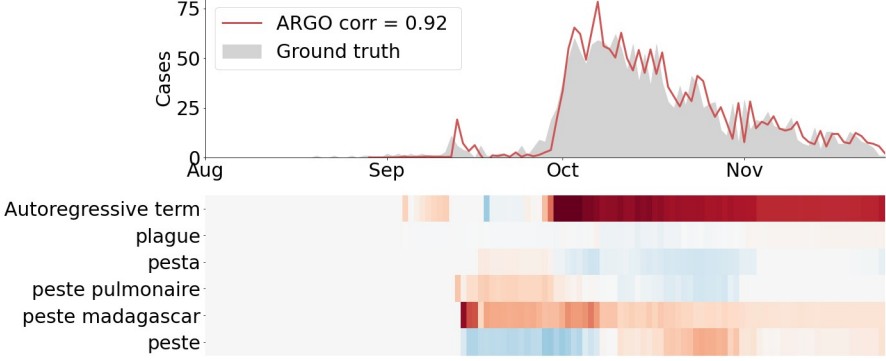

**Fig 3. Evaluating feature importances (coefficients in linear regression) in ARGO for nowcasting plague in Madagascar assuming a reporting delay of one week.** In the heatmap, darkest reds correspond to largest positive coefficients and darkest blues correspond to largest negative coefficients; grey indicates a coefficient of zero. Since the model is trained dynamically, feature importances shift from week to week. Note that the autoregressive term is extremely important, but information from Google search trends is also used, particularly early on in the outbreak.

**Table 2. Comparison of cases reported in epidemiological bulletins to projections produced by our nowcasting models in real-time.** Accuracy is measured in comparison to the ground-truth case counts eventually reported at the end of each outbreak. Accuracy of each model is calculated separately for the second-to-last week with epidemiological data reported in each bulletin, the last week with epidemiological data reported, and the first week without any epidemiological data reported (we do not evaluate longer time horizons since in several outbreaks epidemiological bulletins are produced near-weekly, rendering projections based on previous bulletins obsolete). Accuracy measures are then averaged over epidemiological bulletins for each outbreak (ranging from seven for Zika in Colombia to 17 for Ebola in the DRC), with standard deviations shown in parentheses. The accuracy of the data source producing the most accurate point estimates on average is bolded.

| | Percent error | | | Absolute error | | |
|---|---|---|---|---|---|---|
| | Second-to-last week with epi data reported | Last week with epi data reported | First week without epi data reported | Second-to-last week with epi data reported | Last week with epi data reported | First week without epi data reported |
| *Yellow Fever in Angola (N = 11)* | | | | | | |
| Report | **67.2 (24.9)** | **80.6 (34.4)** | — | 12.8 (9.8) | 17.0 (15.1) | — |
| AR | 124.4 (189.8) | 610.1 (1035.2) | 958.8 (1543.4) | 9.0 (5.4) | 16.4 (12.6) | 31.0 (11.8) |
| GT | 156.5 (286.7) | 397.1 (715.4) | **430.9 (761.5)** | 11.1 (5.1) | **14.4 (7.0)** | **13.4 (7.7)** |
| ARGO | 105.4 (180.1) | 419.9 (720.5) | 540.7 (901.4) | 8.5 (6.1) | 16.4 (7.8) | 18.5 (10.9) |
| *Zika in Colombia (N = 7)* | | | | | | |
| Report | 21.7 (27.4) | 33.2 (8.8) | — | 418.0 (377.2) | 872.2 (182.0) | — |
| AR | 34.0 (46.2) | 38.1 (26.2) | 31.3 (34.7) | 710.3 (529.8) | 1104.4 (1088.2) | 1088.8 (1622.3) |
| GT | 36.9 (14.2) | 31.6 (14.6) | 39.3 (10.7) | 961.4 (316.6) | 962.8 (611.4) | 1214.7 (603.0) |
| ARGO | **20.9 (39.0)** | **14.9 (13.9)** | **15.6 (18.1)** | **344.4 (416.0)** | **334.0 (198.3)** | **584.3 (809.0)** |
| *Ebola in the DRC (N = 17)* | | | | | | |
| Report | **41.8 (31.5)** | 82.5 (20.2) | — | **11.1 (10.3)** | 21.2 (9.4) | — |
| AR | 46.1 (23.3) | **50.2 (20.2)** | **54.5 (31.1)** | 13.3 (10.7) | **14.1 (7.5)** | **18.1 (12.4)** |
| GT | 53.2 (21.5) | 50.8 (25.2) | 58.4 (23.4) | 16.8 (12.3) | 15.8 (12.2) | 19.8 (12.9) |
| ARGO | 43.9 (25.7) | 50.6 (19.2) | 59.6 (26.7) | 13.3 (11.1) | 14.2 (7.4) | 19.5 (11.7) |
| *Plague in Madagascar (N = 12)* | | | | | | |
| Report | **11.6 (14.1)** | **14.9 (17.2)** | — | **2.6 (3.3)** | **5.2 (7.1)** | — |
| AR | 63.3 (72.0) | 40.7 (23.0) | 48.6 (42.1) | 10.0 (5.1) | 8.6 (4.9) | 9.4 (8.3) |
| GT | 65.4 (57.6) | 46.4 (32.6) | 56.9 (49.7) | 13.0 (13.9) | 12.2 (12.0) | 12.4 (12.9) |
| ARGO | 67.7 (76.0) | 38.5 (26.8) | **40.6 (37.1)** | 10.6 (5.6) | 7.3 (4.2) | **7.7 (7.0)** |
| *Cholera in Yemen (N = 12)* | | | | | | |
| Report | **5.1 (6.2)** | 18.9 (9.0) | — | **2045.1 (3273.2)** | 6773.2 (4881.6) | — |
| AR | 9.3 (5.6) | **12.4 (7.0)** | **20.7 (18.3)** | 3160.3 (2252.2) | **3516.3 (1289.1)** | **4739.7 (3236.1)** |
| GT | 49.9 (22.7) | 48.0 (23.8) | 46.6 (23.5) | 18657.1 (11130.5) | 17618.5 (11182.9) | 16013.7 (10839.1) |
| ARGO | 9.3 (5.6) | **12.4 (7.0)** | **20.7 (18.3)** | 3160.3 (2252.2) | **3516.3 (1289.1)** | **4739.7 (3236.1)** |

were released in real-time (Fig 3). In contrast to the first approach, here models were trained on a potentially (and frequently) unreliable ground truth, since future revisions of past disease activity may continually update case reports that are released at any given point. We assessed the feasibility of these models in achieving an estimate of disease activity when there are no epidemiological data available in real-time. This analysis was performed on all 7-17 reports for each of the five disease outbreaks; a selection of case studies are presented here and full charts are included in S6–S10 Figs. In addition, Table 2 compares aggregate measures of the accuracy of each nowcasting model for predicting case counts in real-time alongside the accuracy of the epidemiological reports that were released in real-time.

As shown in Fig 4, we observed that, even in these realistic circumstances, ARGO produced meaningful disease activity estimates that filled the temporal gap introduced by delayed availability of epidemiological reports. The value of Google search data and nowcast models like ARGO is most apparent in outbreaks with long delays between epidemiological updates: consider the upper-right panel of Fig 4, in which ARGO projections provide useful information in the absence of any up-to-date epidemiological data.

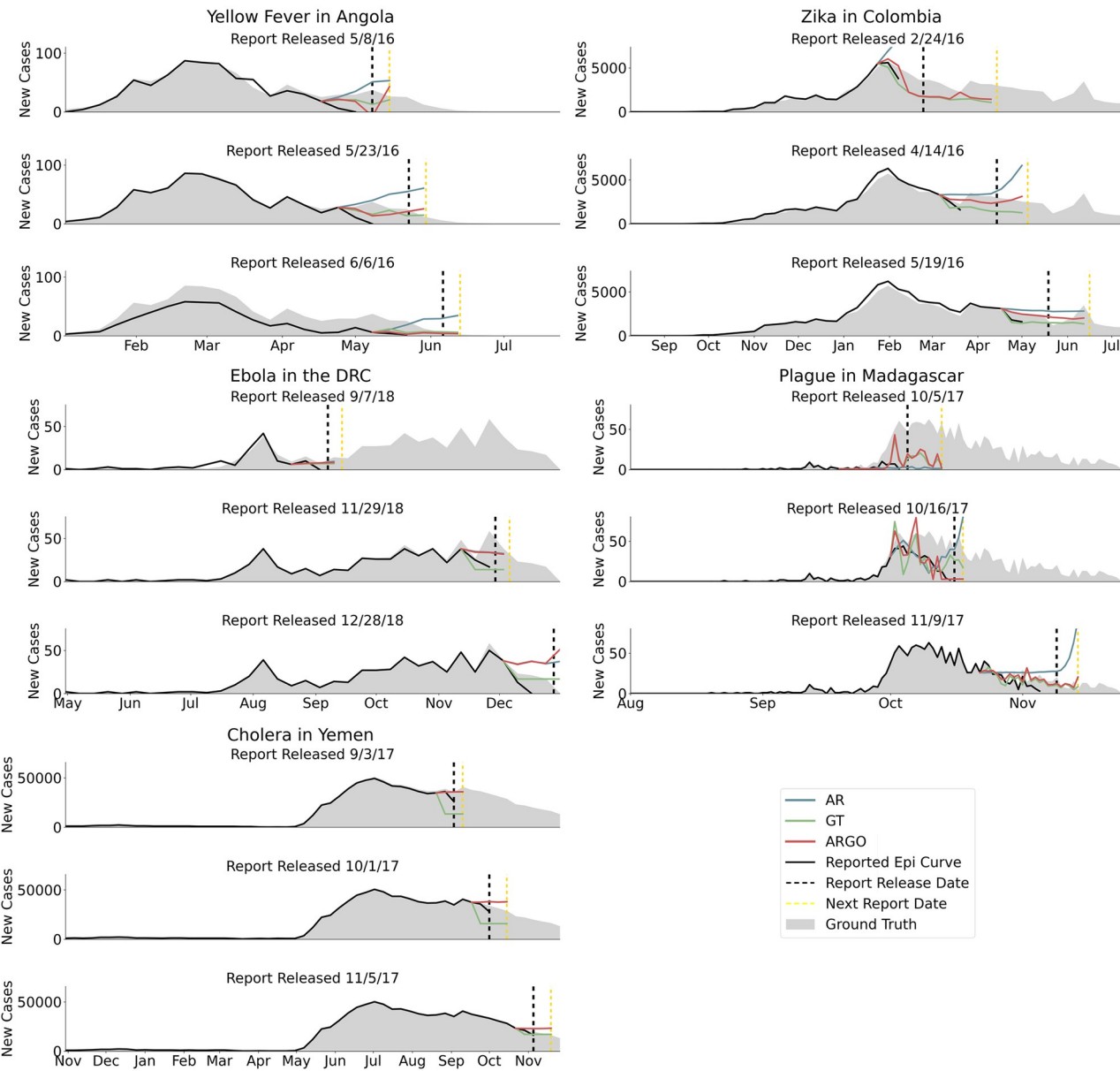

**Fig 4. Summary of evaluation approach based on publicly released reports.** In each figure, the grey filled area is the ground truth data (available at, or after, the end of the outbreak). The black line shows surveillance data released in each report at the time of publication (the date of publication is denoted by the dashed black line), and the colored lines show the real-time predictions of our three models. Here, figures are included for three epidemiological situation reports for each outbreak; more plots with the same evaluation task can be found in S6–S10 Figs. For inclusion in this figure, we chose three evenly-spaced situation reports from the middle of each outbreak.

Further, as shown in Table 2, ARGO was as or more accurate when compared to other models for estimating case counts in real-time for most outbreaks (with the exception of Ebola in the DRC, for which AR was most accurate, and yellow fever in Angola, for which ARGO and GT traded off the position of most accurate). Our models were frequently more reliable than the epidemiological data recorded in the final week in each epidemiological report (for yellow fever in Angola, Zika in Colombia, Ebola in the DRC, and cholera in Yemen, based on absolute error), and occasionally more accurate than data recorded in the second-to-last week

in each epidemiological report (for yellow fever in Angola and Zika in Colombia, based on absolute error). These results suggest that even when epidemiological data are available in near-real-time, they can be complemented by nowcast models which do not suffer from issues of under-reporting.

## Discussion

We show that machine learning techniques that combine real-time disease-related Google search activity with (delayed and frequently incomplete) epidemiological information available during emerging outbreaks can provide useful real-time insights on the likely trajectory of disease transmission. By assessing model predictions in (i) a setting that assumes the continuous availability of delayed epidemiological information (reporting delays of 1-2 weeks with no case revision) and (ii) a set of realistic historical settings where delayed information was unavailable or unreliable (reporting delays of variable week lengths and with case revisions in subsequent epidemiological reports), we demonstrate that incorporating disease-related Google search information improves predictions across several disparate disease and country contexts.

In particular, we demonstrate, for the first time, how a digital nowcast model like ARGO would be deployed in real-time during multiple distinct emerging disease outbreaks with reporting delays and surveillance revisions. We show specifically the insights that would have been accessible in real-time should our approaches have been implemented during the emergence of these outbreaks. Consider, for example, the real-time disease predictions for the 2017 plague outbreak in Madagascar shown on the right-middle panel in Fig 4. The black line, which indicates the number of known reported cases at the time of release of an epidemiological report (Oct. 16, 2017), suggests a sharp decline in cases in October. By the end of the outbreak, it would become clear that there was no decrease in cases in October (ground truth cases produced at the end of the outbreak are shown in gray shading), an insight which was not available in real-time, but which was captured by the Google-based model (GT, green line). We find that the pattern demonstrated in Madagascar generalizes to other diseases and regions: epidemiological reports frequently display a down-turn at the end of the case curve due to under-reporting, implying that the outbreak may be coming to an end. Since our models do not suffer from the under-reporting issue, they exhibit no such downturn, frequently suggesting that the outbreak is ongoing when the most up-to-date epidemiological data suggest otherwise. Moreover, as shown in Fig 4, predictions generated by our models could be used to fill temporal "data gaps" when up-to-date epidemiological data is unavailable.

In addition to showing the potential utility of real-time predictions trained on unreliable or incomplete epidemiological data, our analysis confirms the findings of other digital epidemiological studies that demonstrate the added value of combining Google-based predictions with autoregressive case information [13, 16, 18, 30]. Indeed, the ARGO coefficient heatmaps in Fig 3, S1–S4 Figs reveal that the epidemiological case information from previous weeks has consistently strong predictive power over the course of the outbreak, while the importance of Google predictors fluctuates over time and appears to be most useful in the earlier stages of the studied outbreaks. The phenomenon that past cases are intrinsically linked to future cases is a common feature of infectious disease outbreaks: here, we leverage this fact to improve the accuracy of our predictions, evidenced by the fact that ARGO generally outperforms the Google-only and autoregressive models across diseases and prediction horizons.

Further, our findings suggest that the relative feature importance of autoregressive information and Google search data is dependent on the timescale of disease transmission (serial interval). Specifically, we find that search data appears to posses greater predictive power in diseases with short serial intervals like influenza, and less predictive power in diseases like

Cholera, where transmission time-scales are typically longer. We hypothesize that in diseases that spread quickly and affect large swaths of the population, there is in general more data—both ground truth epidemiological data and trace data from Internet searches—available, so there is a higher signal to noise ratio, and models are better able to generalize from the multiple data streams. Diseases with longer serial intervals, which spread more slowly, will naturally result in scarcer data from all sources.

While there are many promises of using Google data to track and predict outbreaks, there are several limitations to using Google data for epidemiological purposes. In the context of emerging outbreaks, these include bias in the sample of Google users, bias due to search term selection, and bias introduced as a result of media coverage.

Google users are a non-random sub-sample of the population, and this bias is particularly significant in the context of most emerging outbreaks, which occur in developing regions where Internet penetration is relatively low and in which there are significant rich-poor and urban-rural divides in Internet access. As a result, it is possible that much of the disease-related Google search activity may occur in a country's capital, while cases of the disease may occur all over the country or in a specific region with low Internet penetration. Exploration of Google search activity on sub-national levels could help provide insight into this issue, though this bias will likely become less relevant as global Internet penetration in rural regions increases. Relatedly, not all Internet-users are Google users; cultural relevance is an important factor in determining which Internet-based data sources are appropriate for digital epidemiological studies of emerging outbreaks.

Search term selection introduces another form of bias in modeling disease trajectories with search query data. As discussed in more detail in the Materials and Methods section, it is standard in the literature on digital epidemiology to select search terms by mining correlations between search query volumes and epidemiological data during a training period which is disjoint from the period of model evaluation [13, 18, 30]. In emerging outbreaks, however, there is little time for such calibration, so a "common sense" approach to selecting intuitive search terms like the one used in this paper may be more appropriate. Future work could examine the types of search terms that are particularly relevant during outbreaks, perhaps examining temporal heterogeneity in search term frequency to understand how the behavior of searchers changes over the course of an outbreak.

Finally, media coverage may confound the interpretation of our models. In using Google query volumes as a proxy for disease activity, it may be the case that queries come from individuals who are infected or suspect infection. However, Google query volumes inevitably also contain signals resulting from high media coverage (often pervasive during novel and unexpected outbreaks), which prompts large numbers of people in the affected country to search for disease-related terms out of curiosity, seeking news articles. Consider the graph of search volumes for the term "peste" (French for "plague") in Madagascar in Fig 1: there is a sharp spike in volumes in mid-October, which appears anomalous to the incidence curve. It is very reasonable to hypothesize that this spike is the result of the first media coverage of that outbreak.

To evaluate how media coverage may skew Google search volumes, we qualitatively compare signals in Google searches and news report volumes with epidemiological time series. Fig 5 compares the volume of news articles (obtained from the GDELT Global Knowledge Graph [31]), Google search trends, and reported cases side by side for each outbreak. Based on this analysis, it is plausible that ARGO's weaker performance on Ebola in the DRC and on Cholera in Yemen are caused by premature spikes in Google searches. These premature spikes are correlated with early spikes in news coverage, and these early spikes are not found for the

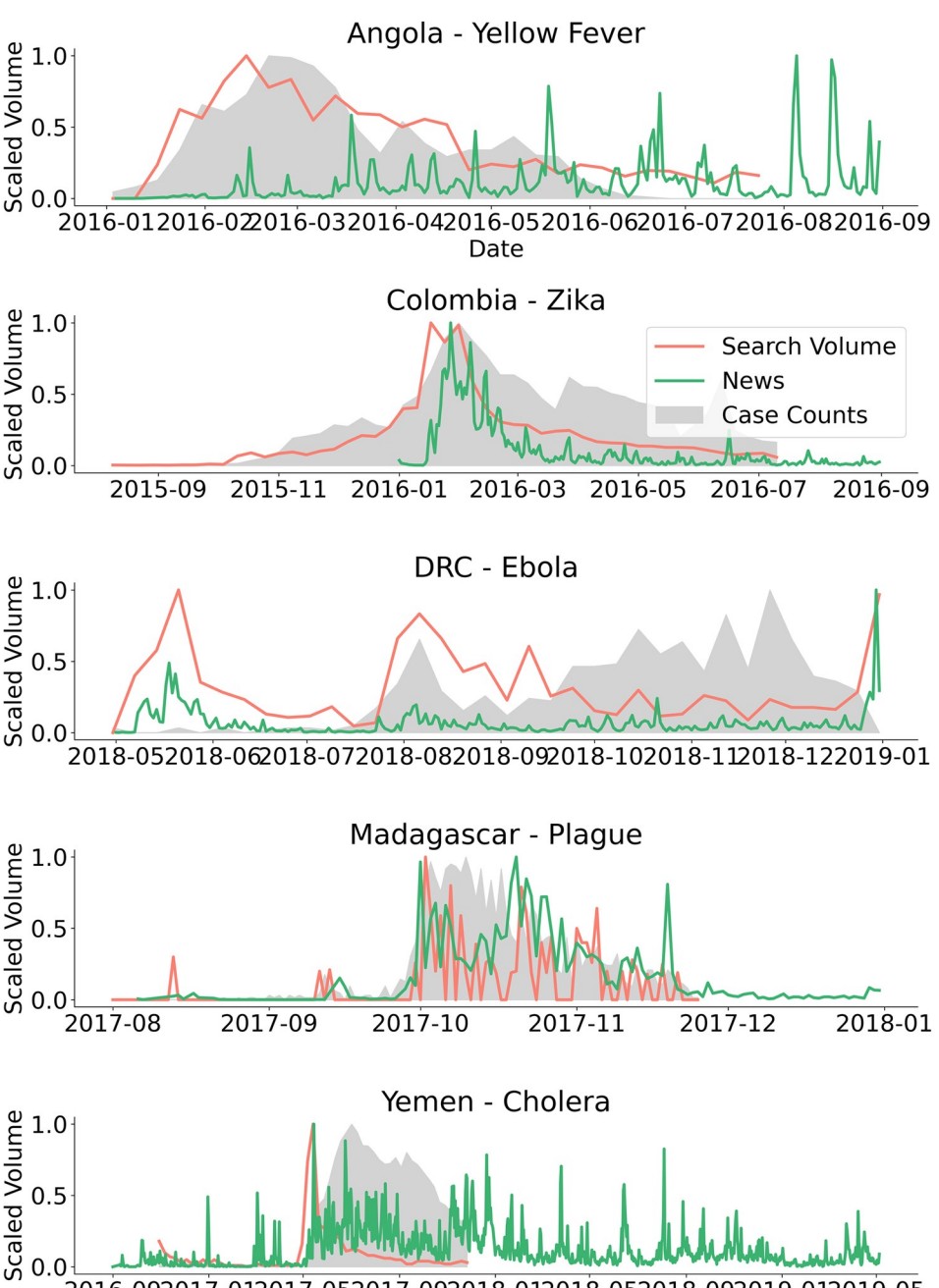

**Fig 5. Comparison of signals in ground-truth epidemiological data, Google search query volumes, and news alerts data from the GDELT Global Knowledge Graph.** Note how media coverage (as captured in the news alerts time-series) may bias predictions based on the GT data. Search term signals are drawn from the same search term as in Fig 1.

other outbreaks where ARGO had better performance. It is likely that hype caused by media coverage biases predictions based on Google search volumes in these analyses.

A final limitation of the work presented here is the use of "ground truth" epidemiological data. We assume that the final epidemiological situation report released for each outbreak we analyze represents the true timing and volume of cases in that outbreak, but in reality it is likely that many cases go unrecorded or misrecorded [1]. Our work points to the need for investments in standard methods of epidemiological data collection, in addition to the potential to complement these standard sources with nontraditional data. Earlier release of case reports—even incomplete ones—could improve model accuracy and provide sounder empirical basis for public health decision making.

Here we have shown how Internet-based data streams can be mined to monitor the progression of emerging outbreaks in low-income settings where traditional surveillance may lag substantially or be rendered inaccurate due to backfilling. We have shown that digital epidemiological methods like ARGO perform well for nowcasting plague in Madagascar, yellow fever in Angola, and Zika in Colombia, but are less effective at tracking cholera in Yemen and Ebola in the DRC. The poor performance for the Ebola and cholera outbreaks could be linked to a combination of low Internet coverage, intense response to news alerts, and rapid shifts in disease dynamics due to population unrest and violence.

We suggest two main directions for future work in this space. First, previous studies have shown that multi-prong approaches based on a variety of traditional non-traditional data sources (including epidemiological data, search data, social media data, news data, electronic health records, and more) are superior to relying on just one or two proxy data sources [14, 15, 30]. Future work could assess the non-traditional and Internet-based data sources available in the settings of novel outbreaks, and consider multi-pronged approaches to tracking outbreaks with multiple novel data streams. Given the results on media skew from this study, models incorporating data on news coverage volume alongside searches and epidemiological data are of particular interest. A second line of future work should focus on the pathogen and population conditions (digital coverage, symptoms specificity, serial interval, mode of transmission, behavior changes, and health interventions) that can make or break digital surveillance in low-income settings, and how to adjust digital surveillance signals for intense media coverage and other exogenous forces.

## Materials and methods

### Data sources

We digitized daily or weekly national case counts from epidemiological situation reports for outbreaks of yellow fever in Angola (Jan. 3—July 31, 2016), Zika in Colombia (Aug. 9, 2015—July 10, 2016), Ebola in the Democratic Republic of the Congo (April 30—Dec. 31, 2018), pneumonic plague in Madagascar (Aug. 1—Nov. 2016), and cholera in Yemen (Oct. 30, 2016—Nov. 26, 2017). We also downloaded country-specific time-series of Google query volumes from the Google Trends API for the same time periods.

**Epidemiological data.** Table 3 summarizes the sources of epidemiological data and key descriptive statistics on the epidemiological dataset for each of the five outbreaks analyzed. For each dataset, we consider the final epidemiological report to be the "ground truth" recording the true onset date for each of the cases in the outbreak; the earlier reports are considered estimates and subject to revision. Note that this assumption requires a larger leap for Ebola and cholera than for the other outbreaks analyzed, as these outbreaks were ongoing at the time of data collection whereas the other outbreaks were completed. Finally, note that, due to issues of data availability, in certain outbreaks the dataset consists of only laboratory-confirmed cases,

**Table 3. Epidemiological data sources.**

| Outbreak | Time period | Temporal granularity | Total cases | Reports | Source |
|---|---|---|---|---|---|
| Yellow Fever in Angola | Jan. 3—July 31, 2016 | Weekly | 879 (confirmed) | 11 | Digitized from plots in PDF situation reports released by WHO [5] |
| Zika in Colombia | Aug. 9, 2015—July 10, 2016 | Weekly | 91,156 (suspected) | 7 | Digitized from plots in PDF epidemiological updates published after Feb. 17 (only updates with Colombia-specific data are included) [6] |
| Ebola in the DRC | Apr. 30—Dec. 31, 2018 | Weekly | 628 (suspected) | 17 | Digitized from plots in PDF situation reports released by the WHO [7] |
| Pneumonic plague in Madagascar | Aug. 1—Nov. 25, 2016 | Daily | 1,857 (confirmed) | 12 | Digitized from plots in PDF situation reports released by the IPM [9] and WHO [8] (only reports containing case counts specifically for pneumonic plague are included) |
| Cholera in Yemen | Oct. 30, 2016—Nov. 26, 2017 | Weekly | 973,802 (suspected) | 13 | Digitized from plots in PDF situation reports released by WHO AFRO [10] |

while in other outbreaks the dataset contains both confirmed and probable (or suspected) cases.

**Google search trends data.** Time-series downloaded from Google search trends [32] describe the number of people searching for a specific keyword, in a specified geographic region, each day, week, or month (normalized to a 0—100 range). Google search trends data was extracted for each outbreak for the same time period as the epidemiological data, on the same temporal granularity as the epidemiological data, and limited to searches in the country of the outbreak. To avoid forward-looking bias, it is standard to select keywords by using Google correlate to find search terms that correlate well with the epidemiological time-series in a training period (which is then not included in the evaluation period) [13, 18, 30]. However, since Google correlate data is not available for any of the countries we analyze, we select a few simple keywords for each outbreak that are clearly related to the disease in question. In certain cases, there is not enough Google search information to yield meaningful results in the sample available through Google search trends: for example, we identified "fièvre hémorragique" and "fievre hemorragique" as relevant search terms for Ebola in the DRC, but were unable to include them due to a lack of available search signal. Similarly, we experimented with including "diarrhea" and the Arabic versions of "cholera" and "diarrhea" for the outbreak of cholera in Yemen, but did not find an improvement in signal over using only "cholera" in English. All search terms used to model each outbreak are listed in Table 4.

**News alert data.** News alert data was obtained from the GDELT Global Knowledge Graph in the form of fractions of daily raw article counts that are relevant to a query. GDELT is a large and regularly updated open database and platform that monitors the world's news media in over 100 languages [31].

**Table 4. Search terms by outbreak.**

| Outbreak | Search terms |
|---|---|
| Yellow Fever in Angola | 'yellow fever', 'febre amarela' |
| Zika in Colombia | 'zika', 'zika sintomas', 'el zika', 'sintomas del zika', 'virus zika', 'zika colombia', 'el zika sintomas', 'el sica' |
| Ebola in the DRC | 'ebola' |
| Plague in Madagascar | 'plague', 'pesta', 'peste', 'peste pulmonaire', 'peste madagascar' |
| Cholera in Yemen | 'cholera' |

## Models

We explored three simple data-driven nowcasting models, emphasizing model simplicity as there is often not enough data available in emerging outbreaks to train a more complex model.

**Linear autoregression (AR).** An autoregressive model uses a linear combination of past observations of disease incidence ("autoregressive terms," $y_{t-i}$) to provide an estimate for synchronous incidence $y_t$. Here, we choose for simplicity to use only the single most recently observed autoregressive term, so the linear autoregression is a univariate linear regression:

$$y_t = \beta y_{t-h} + \alpha \tag{1}$$

The linear regression is optimized over available training observations to minimize mean squared error loss. The time horizon of prediction $h$ depends on the reporting delay in each outbreak; for instance, if there is a two-week reporting delay in a surveillance system, the autoregressive term will be the 2-week lag, so $h = 2$.

**Regression on Google query volumes (GT).** Our second model is a multivariate regression mapping synchronous data on a set of Google query volumes for the set of selected search terms $G = \{g_1, \ldots, g_k\}$ to estimated synchronous incidence. Depending on the number of search terms selected for each outbreak, this regression contains 1-8 variables.

$$y_t = \Sigma_{g \in G} \beta_g g + \alpha \tag{2}$$

We adopt a L1 regularization to prevent overfitting and provide automatic feature selection, with the regularization parameter selected via 5-fold cross validation on the training set from $\{10^{-5}, 10^{-4}, 10^{-3}, 10^{-2}, 10^{-1}\}$. The LASSO regression is optimized over available training observations to minimize mean squared error loss.

**Autoregression and regression on Google query volumes (ARGO).** ARGO combines the AR and GT methods in a single multivariate regression including both a single autoregressive term (the most recently observed incidence value) and a set of synchronous Google query volumes.

$$y_t = \beta y_{t-h} + \Sigma_{g \in G} \beta_g g + \alpha \tag{3}$$

As in GT, ARGO is made more robust with L1 regularization, with the regularization parameter selected via 5-fold cross validation on the training set from $\{10^{-5}, 10^{-4}, 10^{-3}, 10^{-2}, 10^{-1}\}$. The ARGO method used here is a somewhat simplified version of the linear regression on autoregressive data and synchronous Google query data originally developed to nowcast influenza in the United States [13]. When interpreting LASSO coefficients for ARGO, we average coefficients over ten model runs with ten different random seeds to ensure coefficient stability. In practice, we observe that coefficient sizes do not change from run to run.

## Evaluation

We had access only to publicly released epidemiological situation reports, which are typically released somewhat sporadically, exhibiting long reporting delays and gaps where no information is available at all. We assumed that the final situation report released for each outbreak was the "ground truth," recording the true timing and volume of new cases. To capture two possible data-access scenarios, (1) an ideal scenario in which final case numbers are reported 1-2 weeks after they occur, and (2) a more realistic scenario in which case numbers are reported with some delay and possibly corrected at a later date, we adopted two separate methods of evaluation. The first evaluation method assumes a continuous flow of correct epidemiological data and a set reporting delay of one to two weeks. The second method reflects the

reality of many epidemiological reporting systems by using the data presented in publicly released epidemiological reports.

**Evaluation assuming continuous flow of epidemiological data.** The first form of evaluation uses only a single time-series of epidemiological data; the "ground truth" (taken as the last epidemiological report on the outbreak publicly released). We assumed a $h$-week reporting delay and experiment with $h$ taking on values of 1 and 2. Thus this evaluation method represents a near-ideal data access scenario in which case counts, once reported, are never adjusted or corrected. We adopted dynamic training (also known as online learning or walk-forward validation) so that, when predicting each week's incidence, each of the models is trained on all the data available up to that week. Models were then evaluated over the entire time-series based on Pearson's Correlation Coefficient (CORR), root-mean-square error (RMSE), and relative root-mean-square error (rRMSE). Previous work has evaluated using shorter training windows or weighting recent data more heavily in training to focus models on recent disease dynamics [13]. In this case, we elected to use all available data in training due to the short length of each time-series and resulting data scarcity.

$$CORR = \frac{\sum_{i=1}^{n}(y_i - \bar{y})(x_i - \bar{x})}{\sqrt{\sum_{i=1}^{n}(y_i - \bar{y})^2}\sqrt{\sum_{i=1}^{n}(x_i - \bar{x})^2}} \tag{4}$$

$$RMSE = \sqrt{\frac{1}{n}\sum_{i=1}^{n}(y_i - x_i)^2} \tag{5}$$

$$rRMSE = \frac{\sqrt{\frac{1}{n}\sum_{i=1}^{n}(y_i - x_i)^2}}{\bar{y}} \tag{6}$$

**Evaluation based on publicly released epidemiological situation reports.** The ideal data-access scenario described above is not always the case in emerging outbreaks, which are characterized by reporting gaps and revisions of case counts after initial publication. The second method of evaluation recognizes this challenge, and compares the accuracy and timeliness of epidemiological reports that were publicized in each outbreak with the accuracy and timeliness of our three digital epidemiological models. We first empirically estimated the average reporting delay for each outbreak as the average number of days or weeks from initial reporting to a stable count of cases for a given day or week of the outbreak in the epidemiological reports. To account for small human errors in reporting and digitization of reports, we defined a "stable" case count as one that does not change by more than 1% from one week to the next. In practice, we observed a 2-week reporting delay for all five outbreaks presented. Note that while this empirical method requires several weeks of published epidemiological reports, a healthcare system's reporting delay could likely be estimated a priori by its managers.

For each report released during each outbreak, we trained the three listed digital epidemiological models on the data that was stable in the report (according to the calculated reporting delay). We trained models for every time horizon between when stable data in the report ceased to be available and when the next epidemiological report was posted (as a way to evaluate what utility digital epidemiological models would have had at the time). We assume that minimal time would be required for data entry or processing, so our models would be available more or less instantaneously upon the release of an epidemiological report.

In addition to generating and presenting the predictions for each model for each report released in each outbreak, we evaluated the accuracy of each model, on average, in comparison to the accuracy of epidemiological data reported in real time. Specifically, we calculated the absolute and percentage error (in comparison to ground-truth case data) for case counts reported in epidemiological bulletins in the second-to-last and last week of each bulletin:

$$Absolute\ Error = |y_i - x_i| \tag{7}$$

$$Percentage\ Error = \frac{|y_i - x_i|}{y_i} \tag{8}$$

We use absolute and percentage error here—rather than correlation and RMSE as in the first evaluation method—since errors are evaluated at single points in time rather than over the whole of a time-series. We then averaged these second-to-last-week and last-week errors across situation reports for each outbreak (ranging from seven situation reports for the Zika outbreak in Colombia to 17 case reports for the Ebola outbreak in the DRC). In the same way, we evaluated the accuracy of case counts projected by our models for the second-to-last and last week of each bulletin, and the first week after epidemiological data ceased in each bulletin. Again, we averaged the accuracy of each model across case reports for each outbreak, so that mean predictive accuracy could be compared across models and reported surveillance data.

## Supporting information

**S1 Fig. Feature importance heatmaps for nowcasting yellow fever in Angola with ARGO.** In the heatmaps, darkest reds correspond to largest positive coefficients and darkest blues correspond to largest negative coefficients; grey indicates a coefficient of zero.
(TIF)

**S2 Fig. Feature importance heatmaps for nowcasting Zika in Colombia with ARGO.** In the heatmaps, darkest reds correspond to largest positive coefficients and darkest blues correspond to largest negative coefficients; grey indicates a coefficient of zero.
(TIF)

**S3 Fig. Feature importance heatmaps for nowcasting Ebola in the DRC with ARGO.** In the heatmaps, darkest reds correspond to largest positive coefficients and darkest blues correspond to largest negative coefficients; grey indicates a coefficient of zero.
(TIF)

**S4 Fig. Feature importance heatmaps for nowcasting plague in Madagascar with ARGO.** In the heatmaps, darkest reds correspond to largest positive coefficients and darkest blues correspond to largest negative coefficients; grey indicates a coefficient of zero.
(TIF)

**S5 Fig. Feature importance heatmaps for nowcasting cholera in Yemen with ARGO.** In the heatmaps, darkest reds correspond to largest positive coefficients and darkest blues correspond to largest negative coefficients; grey indicates a coefficient of zero.
(TIF)

**S6 Fig. Comparing the accuracy and timeliness of publicly released epidemiological updates from the outbreak of yellow fever in Angola to the accuracy and timeliness of our digital epidemiological models.**
(TIF)

**S7 Fig. Comparing the accuracy and timeliness of publicly released epidemiological updates from the outbreak of Zika in Colombia to the accuracy and timeliness of our digital epidemiological models.**
(TIF)

**S8 Fig. Comparing the accuracy and timeliness of publicly released epidemiological updates from the outbreak of Ebola in the DRC to the accuracy and timeliness of our digital epidemiological models.**
(TIF)

**S9 Fig. Comparing the accuracy and timeliness of publicly released epidemiological updates from the outbreak of plague in Madagascar to the accuracy and timeliness of our digital epidemiological models.**
(TIF)

**S10 Fig. Comparing the accuracy and timeliness of publicly released epidemiological updates from the outbreak of cholera in Yemen to the accuracy and timeliness of our digital epidemiological models.**
(TIF)

## Acknowledgments

This study does not necessarily represent the views of the NIH or the US government.

## Author Contributions

**Conceptualization:** Emily L. Aiken, Sarah F. McGough, Maimuna S. Majumder, Mauricio Santillana.

**Data curation:** Emily L. Aiken, Gal Wachtel, Andre T. Nguyen.

**Formal analysis:** Emily L. Aiken, Andre T. Nguyen.

**Methodology:** Emily L. Aiken, Sarah F. McGough, Gal Wachtel, Mauricio Santillana.

**Project administration:** Mauricio Santillana.

**Supervision:** Cecile Viboud, Mauricio Santillana.

**Validation:** Mauricio Santillana.

**Visualization:** Emily L. Aiken, Andre T. Nguyen.

**Writing – original draft:** Emily L. Aiken, Gal Wachtel.

**Writing – review & editing:** Emily L. Aiken, Sarah F. McGough, Maimuna S. Majumder, Andre T. Nguyen, Cecile Viboud, Mauricio Santillana.

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
