## [Decision Letter · Decision Letter 0]

1 Mar 2020

Dear Ms. Aiken,

Thank you very much for submitting your manuscript "Real-time Estimation of Disease Activity in Emerging Outbreaks using Internet Search Information" for consideration at PLOS Computational Biology.

As with all papers reviewed by the journal, your manuscript was reviewed by members of the editorial board and by several independent reviewers. In light of the reviews (below this email), we would like to invite the resubmission of a significantly-revised version that takes into account the reviewers' comments.

Please consider all feedback from the reviewers to improve the paper, with particular attention to the following points:

Reviewer 2 had two major comments that should be addressed - the first with regard to quantification of performance based on publicly released reports, and the second a more technical critique regarding the methods for estimating coefficients with limited data.Reviewers 1 and 3 have both requested that more guidance be given about when the methods are likely to be reliable. Fleshing out the minimum use case, as suggested by Reviewer 3, will be helpful here, but also consider including some discussion of what would be considered a 'strong' use case.

We cannot make any decision about publication until we have seen the revised manuscript and your response to the reviewers' comments. Your revised manuscript is also likely to be sent to reviewers for further evaluation.

Sincerely,

Juliet R.C. Pulliam, PhD

Guest Editor

PLOS Computational Biology

Virginia Pitzer

Deputy Editor

PLOS Computational Biology

Reviewer's Responses to Questions

**Comments to the Authors:**

Reviewer #1: In this work, the authors predict the short-term trajectory ("nowcasting") of infectious disease outbreaks using regression approaches that consider case reports from earlier in the outbreak, as well as country-specific Internet search activity as provided by Google Trends. Because epidemiological reports are often delayed and sometimes substantially later revised, it makes a lot of sense that we should be leveraging other kinds of information to estimate what is actually happening on the ground. This paper builds on past work that considered more predictable epidemics like seasonal influenza, now applying these methods to harder settings, where analogous past outbreaks may not exist and local political or social unrest may make transmission dynamics both complex and harder to ascertain. As test cases, they chose outbreaks that represent a good challenge to such an effort--they are diverse geographically, in mode of transmission, pathogenicity, etc. Broadly speaking, the paper is well-written and the methods seem sound. I also took a look at the github repository and the IPython notebooks the authors provided, and I appreciate that clarity and transparency therein.

I hope the authors will find my suggestions and questions helpful in further improving the manuscript.

Detailed comments:

Throughout: Zika and Ebola are conventionally capitalized as they are named after places, but yellow fever, plague and cholera are not.

Line 75-76: At this point in the manuscript, it is not clear what is meant by "historic." I initially assumed that some kind of past outbreak data was used to inform a statistical model, but it seems the authors mean strictly data from earlier in the same outbreak.

Sentence spanning 77-80: A comma is missing between "countries" and "weeks", making it hard to read this sentence correctly.

Line 91: What does "accurate" mean? I doubt we are talking about exact numerical values, and anything else is subjective/requires context.

Lines 100-105 and Discussion: The authors find that their methods vary in effectiveness. I find this understandable, but practically speaking it is problematic. How should a policy maker determine whether a nowcast is reasonable to use? Do the authors have any suggestions about how to predict the reliability of a nowcast, or how to characterize the settings where the methods are sane to apply? For context, I raise this point because the authors are clearly concerned with solving a real problem, and thus consider competing scenarios with high quality, regular surveillance data versus sporadic reports subject to revision (an experimental design choice I commend).

Fig 1, possibly some other figures: I suggest not showing 1.25 on the y-axes, as that's an undefined value for data normalized on [0,1]. Also, these legends are highly redundant--they are all normalized to [0,1], so that should just be in the caption rather than twice in every panel; also, as each panel is labeled with the location, there's no need to state it twice in every legend.

Thought motivated by Fig 1: I don't believe the authors mention this, but it seems likely that populations have a kind of "search fatigue" or saturation effect. Early in an outbreak, there may be a heightened need for information about a disease, but unless looking for news updates, it seems that people would not keep searching for the same information. Relatedly, I would imagine there are meaningful trends in the types of search terms used, where people early on look for general information about a disease, how it's transmitted, etc., and later tend to look for information about symptoms and treatment. Is there any evidence for this kind of structure in search term data?

Methods and Discussion: It should be stated that the final surveillance data here is taken as "truth" (and I don't think there's a serious alternative), although real-world reporting practices may vary in time.

Fig 2: The late August spike in the Ebola/DRC nowcasting is somehow driven by autocorrelations in the AR and ARGO models. I understand why the nowcast peaks lag behind the ground truth data, and why the problem is worse for the 2-week lag than the 1-week lag, but what is it about this August peak that is so problematic for the autocorrelation model? I've looked at the methods, and it is not clear to me why this one part of the nowcasts would be so bad. Do the regressions normalize by current size of the outbreak? An increase from 1 to 3 cases from one week to the next is less meaningful than an increase from 100 to 130, but perhaps the AR for Ebola/DRC is "learning" too much from the 1.5 months of basically nothing at the beginning of that time series. The lag between the ground truth data and the nowcasts seems to be roughly double the reporting delay, but mysteriously only for Ebola/DRC--plague/Madagascar is on a similar scale ([0,~50]), but the predictions are much better. I would appreciate if the authors would dig into this a bit more.

Fig 3, and other figures: There is no legend for the heatmaps, nor is the meaning of the colors stated anywhere in text.

Line 171: I have no idea what "within-range" means. "Meaningful" is subjective, but at least it sounds subjective. Unstated, I think, is that the authors are applying some human intuition for what kind of accuracy would be required for appropriate public health responses. They probably should say that this is what they're doing (or explain what alternative criteria are being applied). Also, I think the word "that" is missing between "estimates" and "filled".

Throughout: The space between "Fig." and the number is too big. If this is Latex, the spaces aren't being escaped properly, e.g. should be "Fig.\\ 3". Same for e.g. "Jan.\\ 3".

Lines 215-217: I do not have any intuition for why GT data would be more useful for diseases with short serial intervals. Do the authors have any idea why this might be the case?

Line 221: result, not results

Throughout: Inconsistent capitalization of Internet.

Line 228: Some explanation of how search terms are chosen should be mentioned here, or earlier in the results. This is a very important issue, and shouldn't be buried in the methods.

Lines 242-250: Can media coverage be used as a predictor (aka feature in ML jargon)? I can understand why the authors might not want to use media attention as a predictor of what's happening on the ground, but it seems like it could be very useful as a correction term for search activity. In other words, an increase in search activity is more meaningful if it does not coincide with an increase in media coverage. A locale-specific relationship between media coverage and search patterns could be determined during non-outbreak periods, or for an unrelated health problem.

Lines 298-301: Is Google universally the preferred search engine? Baidu in China? Is Google what Arabic speakers would principally be using?

AR model in Methods: If I understand correctly, at each time point, a linear regression is constructed using all past data to estimate the coefficients, but only the most recent observation (y_t-h) is used as an input. This means that having a long history of small, noisy values would result in a model that has a beta of roughly 0. If more recent data is not weighted as more informative than observations farther in the past, a dramatic increase might still be treated as uninformative, even after a couple such observations. If I am not understanding this, please clarify. In any case, what is going on with the AR model seems relevant to the Ebola/DRC prediction problems and warrants more discussion.

Fig S2: Why are there more rows in the heatmaps than there are row labels?

References: These need to be cleaned up. There are random spaces in URLs, and inconsistent/incorrect capitalization.

A final thought: As perhaps the most famous (and now aborted) example of nowcasting, Google created Flu Trends and Dengue Trends. It might be appropriate to mention that effort in the introduction.

Reviewer #2: In this work, the authors propose that time series of Google search interest can be used to improve situational awareness during outbreaks of emerging diseases. The problem of delays in traditional surveillance channels is well-known and sorely in need of a solution. Although the use of internet search data is well established in fields such as influenza surveillance in the United States, the proposal here is to use such search data in settings where the disease is unusual and internet access is less common. This proposal is relatively unexplored and potentially important. Further, it seems to me that this manuscript contains interesting supporting evidence of this idea. However, the current manuscript has need of revisions to improve the clarity, rigor, and attention to detail of its methods.

Perhaps my greatest concern is the lack of quantification of the performance of the authors' forecasters in the subsection "Evaluation Based on Publicly Released Reports". This is clearly the scenario of greater interest, and although one can see from the figures that the forecasts are reasonable, it is difficult from figure 4 to see that "ARGO appears to most closely estimate the cases that would eventually be reported throughout each outbreak," as the authors write on line 173. A table comparable to Table 1 for this scenario seems like it could provide much clearer support for such a statement. Furthermore, the inclusion of a quantitative metric would allow for later work to easily be compared with this work. On line 381, the authors seem to indicate that such quantification was not possible due to a lack of ground truth data, but the figures invite visual comparison of the forecasts with a ground truth time series, so I find that statement of the authors confusing.

My next greatest concern is the interpretation of the size of the L1-penalized regression coefficients as variable importance in Figure 3 and similar. In my experience, with small data sets such as those analyzed by the authors, the value of these coefficients can be highly sensitive to the random choice of the folds used for cross-validation. Also, the choice of the folds for a time series application where there is a clear temporal correlation structure deserves some discussion. From the code, it seems that the temporal structure of the data is ignored in the choice of folds. How do authors justify this and what effect do they anticipate this has on the selected regularization parameter? Finally, since many of the linear model predictors are likely highly correlated, I think an elastic net penalty would be more appropriate than a lasso penalty if the authors would like to draw some conclusions about the relative importance of the different variables. Zou and Hastie (2005) have shown that the lasso penalty can result in one variable in a correlated group randomly getting a large coefficient, whereas the elastic net penalty can lead to a more equal size of coefficients among members of the group.

I will now list some smaller problems that I noticed when reading the manuscript.

1. Figure 4: The dates on the top of the panels do not always align with the vertical reference lines. For example, consider the panel in the upper left corner.

2. Figure 4 caption: Figs. S9-12 should be Figs. S6-10?

3. Line 215: "we find that GT data appears to posses greater predictive power in diseases with short serial intervals like influenza, and less predictive power in diseases like Cholera, where transmission time-scales are typically longer." The supporting evidence for this statement is unclear.

4. Line 314: "The time horizon of prediction h depends on the reporting delay in each outbreak" It is unclear to me why the authors link the prediction horizon and model lag in this manner. I would consider it simpler to use a lag-1 term in all models and projecting forward multiple steps when necessary for prediction.

5. Equations (2) and (3) make use of variables that the authors never define and do not seem to accurately represent a linear model.

Reviewer #3: I applaud the authors for tacking the difficult problem what do to in the context of inaccurate surveillance reports, particularly for diseases new to specific geographic areas (or new altogether). Current events are only the most recent reminder of how unfortunate delays can be. The paper is relevant, well-written, and the examples provided are comprehensive. Below are some suggestions for improvement.

Lines 171-172: Is there a word missing here? Please check.

The severe delays in getting even preliminary reports out is on painful display in the examples provided (Figure 4). Each epi curve in that figure has a ‘report released’ line and a ‘next report released’ line, but the ‘next report released’ timing does not align with the following epicurve’s ‘report released’ line. I see the supplemental files with all of the reports, but how did the authors choose which to present in the main paper?

Were the authors able to make some kind of quantitative measurement of the about of ‘back corrections’ that appeared in consecutive reports and were the magnitude of ‘misreports’ associated with the model performance?

Lines 196 – 200: The authors propose a minimal case use for these models – that they could at least signal whether the outbreaks are increasing or decreasing, or ‘over’ or ‘not’. Can the authors quantify in the results the number of times each model got this right when the report got it wrong? Some kind of quantification to justify this claim would be helpful. It’s clear that it happened in one case, but some additional evidence that this is a common occurrence would be useful. Also, it’s not clear in the methods, but is the assumption that the forecast would be available on the day that the reports came out? Is there any data entry or analysis time included in the assumptions of when forecasts would be available or is it reasonable to assume these are instantaneous?

The authors should consider making a plug for investments in better traditional epi surveillance, particularly for emerging threats. Models are very helpful, but only once we identify outbreaks and have some data coming in. The findings showing that even incomplete reports could yield useful model results is a strong rationale for releasing data as early as possible. Early data release remains a barrier to protecting global health.

**Have all data underlying the figures and results presented in the manuscript been provided?**

Reviewer #1: Yes

Reviewer #2: Yes

Reviewer #3: Yes

PLOS authors have the option to publish the peer review history of their article (what does this mean?). If published, this will include your full peer review and any attached files.

Reviewer #1: Yes: Thomas J. Hladish

Reviewer #2: Yes: Eamon B. O'Dea

Reviewer #3: No
---

## [Decision Letter · Decision Letter 1]

27 May 2020

Dear Ms. Aiken,

Thank you very much for submitting your revised manuscript "Real-time Estimation of Disease Activity in Emerging Outbreaks using Internet Search Information" for consideration at PLOS Computational Biology, and for your careful response to the previous round of reviews. As with all papers reviewed by the journal, your manuscript was reviewed by members of the editorial board and by several independent reviewers. The reviewers appreciated the attention to an important topic. Based on the review of the revision, we are likely to accept this manuscript for publication, providing that you modify the manuscript according to the review recommendations.

In particular, please make sure that the GitHub repository is updated to include the data underlying the revised figures and summary statistics.

Sincerely,

Juliet R.C. Pulliam, PhD

Guest Editor

PLOS Computational Biology

Virginia Pitzer

Deputy Editor

PLOS Computational Biology

[LINK]

Reviewer's Responses to Questions

**Comments to the Authors:**

Reviewer #2: The reviewers have scrupulously and satisfactorily addressed all my prior concerns. On reviewing the revisions I have just one final concern, which is the level of statistical significance of reported differences in forecasting accuracy. First, I may have missed it, but I did not see the meaning of the parenthesized values in Table 2 of the revised manuscript. I suspect these are standard errors. In the case that they are, it would appear that most differences in forecast accuracy do not have a very high level of statistical significance. Supposing that to be the case, I don't think that is necessarily a reason to remove the claims. My suggestion is simply that the authors should qualify any claims about the method of producing the most accurate forecasts if they cannot provide evidence that it is significantly better than other methods. Alternatively, the authors could make the results stronger by reporting the level of significance of a difference in accuracy if it is non-negligible.

**Have all data underlying the figures and results presented in the manuscript been provided?**

Reviewer #2: No: The GitHub repository the authors refer readers to in "Code Availability" has not been updated since September 2019, therefore it seems unlikely that the underlying data for revised figures and summary statistics are available.

PLOS authors have the option to publish the peer review history of their article (what does this mean?). If published, this will include your full peer review and any attached files.

Reviewer #2: Yes: Eamon B. O'Dea
---

## [Editor Report · Decision Letter 2]

1 Jul 2020

Dear Ms. Aiken,

We are pleased to inform you that your manuscript 'Real-time Estimation of Disease Activity in Emerging Outbreaks using Internet Search Information' has been provisionally accepted for publication in PLOS Computational Biology.

Best regards,

Juliet R.C. Pulliam, PhD

Guest Editor

PLOS Computational Biology

Virginia Pitzer

Deputy Editor

PLOS Computational Biology

---

## [Editor Report · Acceptance letter]

10 Aug 2020

PCOMPBIOL-D-19-02045R2 

Real-time Estimation of Disease Activity in Emerging Outbreaks using Internet Search Information

Dear Dr Aiken,

I am pleased to inform you that your manuscript has been formally accepted for publication in PLOS Computational Biology. Your manuscript is now with our production department and you will be notified of the publication date in due course.

With kind regards,

Sarah Hammond
